# Multidimensional sleep profiles via machine learning and risk of dementia and cardiovascular disease

Clémence Cavaillès [1] ✉, Meredith Wallace[2], Yue Leng [1], Katie L. Stone[3,4], Sonia Ancoli-Israel[5] & Kristine Yaffe[1,6,7]

## Abstract

**Background** Sleep health comprises several dimensions such as sleep duration and fragmentation, circadian activity, and daytime behavior. Yet, most research has focused on individual sleep characteristics. Studies are needed to identify sleep/circadian profiles incorporating multiple dimensions and to assess their associations with adverse health outcomes.

**Methods** This multicenter population-based cohort study identified 24 h actigraphy-based sleep/circadian profiles in 2667 men aged ≥65 years using an unsupervised machine learning approach and investigated their associations with dementia and cardiovascular disease (CVD) incidence over 12 years.

**Results** We identify three distinct profiles: active healthy sleepers (AHS; 64.0%), fragmented poor sleepers (FPS; 14.1%), and long and frequent nappers (LFN; 21.9%). Over the follow-up, compared to AHS, FPS exhibit increased risks of dementia and CVD events (HR = 1.35, 95% CI = 1.02-1.78 and HR = 1.32, 95% CI = 1.08-1.60, respectively) after multivariable adjustment, whereas LFN show a marginal association with increased CVD events risk (HR = 1.16, 95% CI = 0.98-1.37) but not with dementia (HR = 1.09, 95% CI = 0.86-1.38).

**Conclusions** These results highlight potential targets for sleep interventions and the need for more comprehensive screening of poor sleepers for adverse outcomes.

## Plain language summary

Although sleep health encompasses multiple dimensions, most research has evaluated these features in isolation. We aimed to identify multidimensional sleep/circadian profiles and to examine their associations with the incidence of adverse health outcomes. Using wearable device activity data from a prospective cohort of older men, we identified three profiles: active healthy sleepers (AHS), fragmented poor sleepers (FPS), and long and frequent nappers (LFN). Compared to AHS, FPS had increased risks of developing dementia and cardiovascular disease (CVD) events over 12 years whereas LFN tended to have an increased risk of CVD events only. These results suggest potential targets for sleep interventions and underscore the critical need for comprehensive sleep health assessment in clinical practice and research settings.

Growing evidence has linked individual sleep characteristics and disturbed circadian rhythms with adverse health outcomes in older adults, including neurodegenerative and cardiovascular diseases (CVDs), two leading causes of disability and mortality worldwide[1–4]. However, the literature remains inconsistent[5–8]. Some studies have associated both short and long sleep duration with increased dementia risk[9], while others found conflicting associations[7,10,11]. Similarly, although some research has suggested that more frequent or long naps were associated with a higher risk of CVD[4,12], others showed a protective effect[13].

These conflicting findings may be partly due to the lack of consideration of the multidimensional nature of sleep. Research has

primarily examined sleep characteristics in isolation, whereas sleep involves a complex interplay of multiple dimensions such as duration, continuity, quality, circadian rhythmicity, and napping[14]. Adopting a holistic approach by considering common combinations of sleep characteristics could improve our understanding of multidimensional sleep patterns and their associations with outcomes. Moreover, such approach offers key methodological advantages over analyzing sleep characteristics in isolation, such as capturing interactions between multiple sleep dimensions and improving differentiation of outcome risks by leveraging more homogenous groups. Therefore, investigating these associations can provide valuable insights for public health strategies, aiding

[1]Department of Psychiatry and Behavioral Sciences, University of California San Francisco, San Francisco, CA, USA. [2]Department of Psychiatry, University of Pittsburgh, Pittsburgh, PA, USA. [3]Research Institute, California Pacific Medical Center, San Francisco, CA, USA. [4]Department of Epidemiology and Biostatistics, University of California San Francisco, San Francisco, CA, USA. [5]Department of Psychiatry, University of California San Diego, La Jolla, CA, USA. [6]Department of Neurology, University of California San Francisco, San Francisco, CA, USA. [7]Department of Epidemiology, University of California San Francisco, San Francisco, CA, USA. ✉e-mail: clemence.cavailles@inserm.fr

the identification of at-risk populations and targeted treatments or interventions.

In a community-dwelling cohort of older men, our objectives were: (1) to identify actigraphy-derived sleep health profiles based on multi-dimensional objective sleep and rest-activity variables, by using a novel and flexible clustering method; and (2) to investigate the longitudinal associations between these profiles and the incidence of dementia and CVD events over 12 years.

We identify three common sleep/circadian profiles: active healthy sleepers (AHS), fragmented poor sleepers (FPS), and long and frequent nappers (LFN). Compared to AHS, FPS exhibit higher risks of developing dementia and CVD over 12 years whereas LFN show a marginal association with CVD. These findings highlight potential targets for sleep interventions and the need for more comprehensive screening of poor sleep health for adverse outcomes.

## Methods
We followed the Strengthening the Reporting of Observational Studies in Epidemiology (STROBE) reporting guidelines.

### Study design
From 2000 to 2002, the Osteoporotic Fractures in Men Study (MrOS) enrolled 5994 community-dwelling men aged ≥65 years, able to walk without assistance, and without bilateral hip replacements, at six clinical centers across the United States[15,16]. Among them, 3135 were recruited into the ancillary MrOS sleep study and underwent a comprehensive sleep assessment between 2003 and 2005 (our study baseline). Men were screened for use of mechanical devices, including pressure masks for sleep apnea (positive airway pressure or oral appliance devices), or nocturnal oxygen therapy, and were excluded if they reported nightly use of any of these devices (except for intermittent users, $n = 49$). We excluded 331 men with missing actigraphy data or with fewer than 3 days or fewer than 3 nights of actigraphy data, and 137 with significant cognitive impairment at baseline (Modified Mini-Mental State Examination (3MS) score <80 or taking dementia medication), leading to a sample of 2667 participants (Supplementary Fig. 1). All participants provided written informed consent and the study was approved by the Institutional Review Board (IRB) at each site. Our analytic study was approved by University of California San Francisco IRB.

### Actigraphy
Participants wore a SleepWatch-O actigraph (Ambulatory Monitoring, Inc.) continuously on their nondominant wrist for ≥4 consecutive 24 h periods. Data were collected in proportional integration mode and scored by epoch to estimate wake and sleep periods using Action W-2 software and the University of California, San Diego scoring algorithm[17]. Trained scorers at the San Francisco Coordinating Center edited the data using participants' sleep diaries to identify time in and out of bed as well as periods when the interval should be deleted because the watch was removed. Sleep indices were summarized across the monitoring period using means and standard deviations (SDs)[18,19]. Circadian rest-activity rhythm indices were generated using parametric extended cosine models and nonparametric variables[20,21]. A total of 37 actigraphy variables were examined and described in Table 1. There was a median of 5 (range 3–13) nights of actigraphy.

### Dementia incidence
Over 12 years, participants attended four follow-up visits where they reported any physician-diagnosed dementia and their medication use, bringing all medications taken within the past 30 days. Dementia medication use was categorized based on the Iowa Drug Information Service Drug Vocabulary[22]. In addition, trained staff administered the 3MS test to assess global cognitive function. Similarly to previous studies[5,23], incident dementia at any follow-up visit was defined by meeting at least one of the following criteria: (i) self-reported physician-diagnosed dementia; (ii) dementia medication use (verified by clinic staff based on examination of pill bottles); or (iii) a change in 3MS score of ≥1.5 SDs worse than the mean change from

baseline to any follow-up visit. Participants were censored at the date of the diagnostic visit, death, or last visit.

### Cardiovascular disease event incidence
Participants were surveyed for incident CVD events by postcard and phone contact every 4 months for ~12 years, with a response rate over 99%. Relevant medical records and documentation (e.g., laboratory results, diagnostic studies, notes, and discharge summaries) from any potential incident clinical events were obtained by the clinical center. For fatal events, the death certificate and hospital records from the time of death were collected. For fatal events that were not hospitalized, a proxy interview with next of kin and hospital records from the most recent hospitalization in the last 12 months were obtained. These documents were used to determine the underlying cause of death. For both nonfatal and fatal CVD events, all documents were adjudicated by a board-certified cardiologist using a pre-specified adjudication protocol. Inter-rater agreement was periodically evaluated by one or more expert adjudicator(s) in a random subset of events to ensure quality control. Confirmed cardiovascular events were grouped as follows: 1) Coronary heart disease event: acute myocardial infarction (ST or non-ST elevation), sudden coronary heart disease death, coronary artery bypass surgery, mechanical coronary revascularization, hospitalization for unstable angina, ischemic congestive heart failure, or other coronary heart disease event not described above; 2) Cerebrovascular event: stroke or transient ischemic attack; 3) Peripheral vascular disease event: acute arterial occlusion, rupture, dissection, or vascular surgery; and 4) All-cause cardiovascular disease event: combines coronary heart disease, cerebrovascular event, and peripheral vascular disease. Participants were censored at the date of the first CVD event, death, last contact before March 1, 2015, or on March 1, 2015.

### Covariate data
All participants completed questionnaires at baseline, which included items about demographics, living alone, smoking status, caffeine intake (mg/day), and alcohol use (>1 alcoholic drink/week). Level of physical activity was assessed using the Physical Activity Scale for the Elderly[24]. Participants also self-reported their medical history, specifically prior physician diagnosis of heart attack, stroke, diabetes mellitus, and hypertension. The Geriatric Depression Scale was used to assess depressive symptoms, with higher scores corresponding to higher levels of depression[25]. Participants were asked to bring in all medications used within the preceding 30 days. All prescription and nonprescription medications were entered into an electronic database and each medication was matched to its ingredient(s) based on the Iowa Drug Information Service Drug Vocabulary (College of Pharmacy, University of Iowa, Iowa City, IA)[22]. The use of antidepressants, benzodiazepines, and other sleep medications (non-benzodiazepines, non-barbiturate sedative hypnotics) were categorized. A comprehensive examination included measurements of body weight and height. Body mass index (BMI) was calculated as weight in kilograms divided by the square of height in meters. A subset of participants underwent a single-night in-home polysomnography (Compumedics, Safiro, Inc., Melbourne, Australia)[26]. The apnea-hypopnea index (AHI) was calculated as the total number of apneas and hypopneas per hour of sleep (accompanied by a ≥ 3% oxygen desaturation).

### Statistics and reproducibility
We conducted a cluster analysis to identify distinct sleep/circadian profiles. Firstly, we selected variables for inclusion in the analysis, considering the high correlation among the actigraphy variables (Supplementary Fig. 2). When the correlation coefficient between two variables was >0.70, we retained only one of the variables prioritized on clinical meaningfulness, resulting in a final selection of 20 sleep/circadian variables. Secondly, we performed a principal component analysis (PCA) on the 20 selected variables to reduce data dimensionality (while preserving most of the data variation) and enhance the efficacy of subsequent clustering. The 20 variables included in the PCA for cluster creation are highlighted in Table 1.

**Table 1 | Description and interpretation of all actigraphy-derived variables**

| Actigraphy-derived variables | Definition | Interpretation |
|---|---|---|
| Sleep variables | | |
| Bed Time | Mean of Bed Time. | |
| Midpoint (Bed Interval) | Mean of midpoint of Bed to Wake-Up Time. | |
| Midpoint (Onset Interval) | Mean of midpoint of Sleep Onset to Wake-Up Time. | |
| Minutes Napping* | Mean of minutes napping per day, for only naps ≥ 5 min. | |
| Number of Naps | Mean of number of naps per day of duration ≥ 5 min. | |
| Sleep Efficiency | TST / TIB x 100 | Higher value indicates better sleep efficiency. |
| SD Bed Time | Standard deviation of Bed Time. | Higher value indicates poorer rhythmicity. |
| SD Midpoint (Onset interval) | Standard deviation of midpoint Sleep Onset to Wake-Up Time. | Higher value indicates poorer rhythmicity. |
| SD Midpoint Time (Bed interval) | Standard deviation of midpoint of Bed to Wake-Up Time. | Higher value indicates poorer rhythmicity. |
| SD Sleep Onset* | Standard deviation of Sleep Onset Time. | Higher value indicates poorer rhythmicity. |
| SD Wake-Up Time* | Standard deviation of Wake-Up Time. | Higher value indicates poorer rhythmicity. |
| Sleep Latency* | Mean of minutes from Bed to Sleep Onset Time. | |
| Sleep Maintenance | TST / TOW x 100 | Higher value indicates better sleep maintenance. |
| Sleep Onset Time* | Mean of Sleep Onset Time. | |
| Time from Onset to Wake-Up (TOW) | Mean of minutes from Sleep Onset to Wake-Up Time. | |
| Time in Bed* | Mean of minutes from Bed to Wake-Up Time. | |
| Total Sleep Time (TST)* | Mean of minutes of sleep from Bed Time to Wake-Up Time. | |
| Wake After Sleep Onset* | Mean of minutes awake after sleep onset. | Higher value indicates more sleep fragmentation. |
| Wake-Up Time* | Mean of Wake-Up Time. | |
| Circadian activity rhythm variables | | |
| Parameters computed from extended cosine model (ECM) | | |
| Acrophase* | Time of peak (i.e., highest) activity. | Later value indicates later peak of activity and may reflect a more delayed phase. |
| Alpha* | Width of peaks relative to troughs. | Higher value indicates that the peaks are narrow (shorter period of activity) and the troughs are wide (longer period of inactivity/sleep). |
| Amplitude | Peak to nadir difference. | Higher value indicates higher overall rhythmicity. |
| Beta* | Steepness of the rise and fall of the fitted curve. | Higher value (more square-shaped curve) indicates steeper rise and fall and may reflect a more constant level of daytime activity. |
| Down-Mesor | Time of switch from high to low activity (below to above mesor). | Later value indicates later time of declining activity. |
| Mesor* | (Minimum + Amplitude) / 2; mean level of activity. | Higher value indicates higher average level of activity. |
| Minimum* | Minimum value of activity. | Higher value indicates more nighttime activity. |
| Pseudo-F* | Goodness of model fit. | Higher value indicates greater robustness of the RAR and overall rhythmicity. |
| Up-Mesor | Time of switch from low to high activity (above to below mesor). | Later value indicates later time of increasing activity. |
| Nonparametric parameters | | |
| Interdaily stability (IS)* | Consistency of the 24 h RAR between days. | Higher value indicates better consistency of the 24 h RAR between days. |
| Intradaily variability (IV)* | Within-day fragmentation of the 24 h RAR. | Higher value indicates a more fragmented RAR within-day. |
| L5 | Mean activity level during the least active consecutive 5-h. | Higher value indicates less restful sleep. |
| M10 | Mean activity level during the most active daily consecutive 10-h. | Higher value indicates a more active wake period. |
| Midpoint of L5* | Midpoint time of L5. | Indicates whether a person goes to bed earlier or later in the day. |
| Midpoint of M10 | Midpoint time of M10. | Indicates whether a person is most active earlier or later in the day. |
| Start of L5* | Start time of L5. | Indicates the phase of the most restful hours. |
| Start of M10* | Start time of M10. | Indicates the phase of the most active hours. |
| Relative amplitude (RA) | (M10 − L5) / (M10 + L5). | Higher value indicates a more robust 24 h rhythm; reflecting higher activity during wake and relatively lower activity during night. |

* Variables used to create the clusters

The number of principal components (PCs) was determined according to the Kaiser-Guttman criteria[27], considering PCs with eigenvalues > 1, and complemented by visual inspection of the scree plot to identify the elbow point in the eigenvalues distribution (Supplementary Fig. 3)[28]. Thirdly, sleep/circadian profiles were identified using Multiple Coalesced Generalized Hyperbolic Distribution (MCGHD; MixGHD package in R) mixture models based on the six PCs derived from the PCA[29,30]. This method, as opposed to standard clustering approaches, was chosen for its ability to accommodate potentially skewed and/or asymmetric clusters, an important consideration given the skewed distributions often observed in actigraphy data (distributions are described in Supplementary Fig. 4). We explored models comprising one to five clusters, using k-medoids as the starting criterion, and determined the optimal number of clusters by examining the Bayesian Information Criteria (BIC), the Akaike Information Criteria (AIC) and the Integrated Complete-data Likelihood (ICL) (higher values indicating a better fit for the data). The optimal number of clusters was determined by selecting solutions displaying an elbow in the AIC and BIC plots and/or a subsequent drop in ICL. To assess the stability of the clustering solution, we applied a subsampling approach in which 5% of participants were randomly excluded, and the clustering algorithm was rerun on each subsample ($n = 100$). The similarity between each subsample clustering and the original full-sample clustering was quantified using the Adjusted Rand Index (ARI), which ranges from −1 (less agreement than expected by chance between two clusterings) to 1 (perfect agreement) with 0 indicating random agreement.

Sleep and circadian characteristics were compared across clusters using the Kruskal-Wallis test. We calculated effect sizes using the eta-squared ($\eta^2$) statistic to characterize the magnitude of cluster differences. The effect was considered small for $\eta^2 < 0.06$, moderate for $0.06 \leq \eta^2 < 0.14$, and large for $\eta^2 \geq 0.14$[31]. In post-hoc analyses, we performed multiple pairwise-comparisons using Dunn's test adjusted for multiple comparisons with the Bonferroni correction. The sleep and circadian characteristics with the largest effect sizes were presented in radial plots for each sleep profile (Fig. 1). In a sensitivity analysis, clusters were re-determined after excluding participants who intermittently used nightly mechanical devices during sleep ($n = 37$) to assess their influence of the clustering results.

We performed unadjusted and multivariable adjusted Cox proportional hazards models with age as time scale to investigate whether identified sleep profiles were associated with the incidence of dementia and CVD events over 12 years. Covariates were selected based on potential biological plausibility, and included study site, race/ethnicity, education, smoking status, caffeine intake, alcohol use, physical activity, BMI, history of diabetes mellitus and hypertension, depressive symptoms, and sleep-related medications use. We assessed the proportional hazard assumption for each independent variable by examining the Schoenfeld residuals. If a variable violated the assumption, we carried out a stratified Cox regression model for that specific variable.

In sensitivity analyses, models were further adjusted for (i) history of heart attack and stroke, (ii) baseline AHI, (iii) living alone, and (iv) baseline 3MS score (for dementia analysis). We also excluded participants with incident dementia at the first follow-up visit to minimize reverse causation (for dementia analysis), and those with history of heart attack and stroke to minimize confounding bias (for CVD analysis).

Significance level was set at a two-sided $p < 0.05$ and statistical analyses were performed using R version 4.3.0.

## Reporting summary
Further information on research design is available in the Nature Portfolio Reporting Summary linked to this article.

## Results
A total of 2667 men were eligible for cluster analysis. At baseline, participants had a median age of 75 years (interquartile range [IQR] = 72–80), 20.2% had a high school education or lower, and 90.0% were White. Compared to included participants, excluded men ($n = 468$) were older, less educated, more likely to be non-White and to live alone, and had less physical activity and alcohol consumption, but higher depressive symptoms and sleep medication use (Supplementary Table 1).

### Sleep profiles
The PCA conducted on the 20 selected variables resulted in six PCs, collectively explaining 75.6% of the variance (Supplementary Fig. 3). The PCA loadings for each variable across PCs are presented in Supplementary Table 2. After examining the AIC, BIC, and ICL of the MGHD models, three distinct sleep/circadian profiles were identified (Supplementary Fig. 5): active healthy sleepers [AHS; $n = 1,707$ (64.0%)], fragmented poor sleepers [FPS; $n = 376$ (14.1%)], and long and frequent nappers [LFN; $n = 584$ (21.9%)]. The stability of the clustering results, as assessed by the ARI, indicated moderate to good quality (mean ARI = 0.58, median = 0.55, Q1 = 0.50, Q3 = 0.74). The three groups had a median of 5 nights of actigraphy (IQR = 5–5), and their ranges were as following: 3 to 11 for AHS, 4 to 13 for FPS, and 3 to 8 for LFN. All sleep characteristics are described in Table 2.

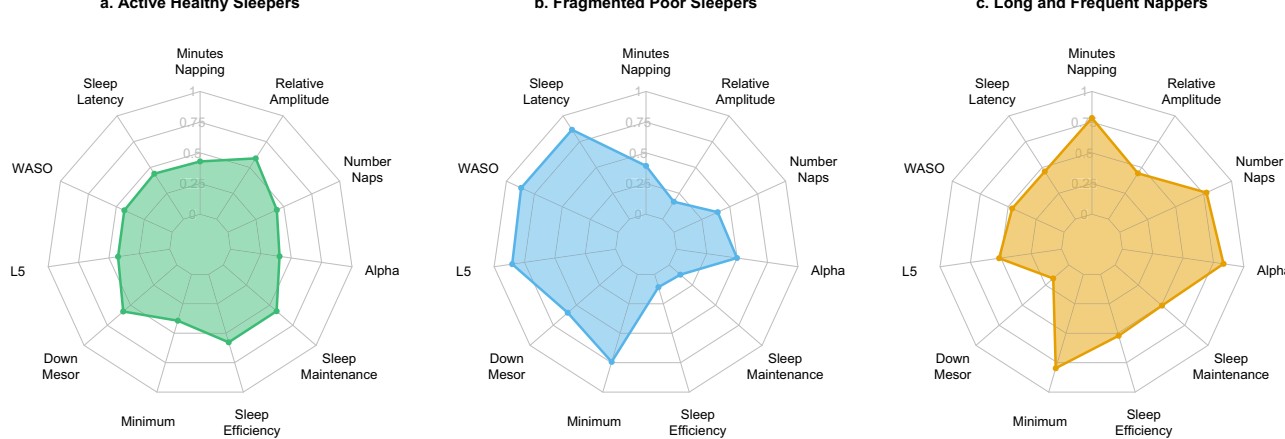

**Fig. 1 | Radial plots displaying the median quantile rankings of sleep and circadian characteristics with large effect sizes for each sleep profile.** Panels (**a**−**c**) display radial plots for the profiles of active healthy sleepers, fragmented poor sleepers, and long and frequent nappers, respectively. The values represent the median quantile rank of each characteristic within the sample, grouped by profile. The sample's highest ranked value is represented by the maximum value of 1, the median ranked value by 0.50, and the lowest ranked value by 0. These values illustrate the relative central distributions across each sleep profile. The sample size was $n = 2667$. Abbreviations: WASO wake after sleep onset

**Table 2 | Sleep characteristics among the 2667 participants according to identified multidimensional sleep clusters**

| Sleep variables | Active Healthy Sleepers ($n$ = 1707) | Fragmented Poor Sleepers ($n$ = 376) | Long and Frequent Nappers ($n$ = 584) | Effect size ($\eta^2$) | $p$-value[c] | Post hoc[d] |
|---|---|---|---|---|---|---|
| | Median (IQR) | Median (IQR) | Median (IQR) | | | |
| Variables used to create the clusters | | | | | | |
| Alpha[a] | −0.40 (−0.50;−0.28) | −0.35 (−0.46;−0.22) | −0.09 (−0.25;0.10) | 0.213 | <2.2e-16 | LFN > FPS > AHS |
| Minimum[a] | 234 (3;385) | 521 (265;756) | 588 (330;781) | 0.193 | <2.2e-16 | FPS,LFN > AHS |
| Wake After Sleep Onset | 61 (42;85) | 126 (97;159) | 65 (43;93) | 0.188 | <2.2e-16 | FPS > AHS,LFN |
| Minutes Napping | 34 (16;60) | 31 (13;53) | 79 (48;127) | 0.160 | <2.2e-16 | LFN > AHS,FPS |
| Sleep Latency | 18 (11;30) | 53 (31;96) | 19 (11;32) | 0.157 | <2.2e-16 | FPS > AHS,LFN |
| SD Sleep Onset | 0.57 (0.37;0.83) | 1.14 (0.80;1.64) | 0.58 (0.37;0.90) | 0.131 | <2.2e-16 | FPS > AHS,LFN |
| Acrophase[a] | 14.38 (13.77;14.98) | 14.69 (13.89;15.57) | 13.51 (12.90;14.16) | 0.114 | <2.2e-16 | FPS > AHS > LFN |
| Total Sleep Duration | 400 (362;438) | 336 (273;395) | 381 (342;424) | 0.074 | <2.2e-16 | AHS > LFN > FPS |
| Pseudo-F[a] | 1078 (781;1421) | 861 (637;1169) | 805 (558;1109) | 0.065 | <2.2e-16 | AHS > FPS > LFN |
| Time in Bed | 489 (453;524) | 530 (482;570) | 475 (440;514) | 0.056 | <2.2e-16 | FPS > AHS > LFN |
| Start of M10[b] | 8.2 (7.3;9.2) | 8.7 (7.7;10.1) | 7.6 (6.9;8.6) | 0.039 | <2.2e-16 | FPS > AHS > LFN |
| Sleep Onset Time | 23.1 (22.4;23.7) | 23.7 (22.7;24.9) | 23.4 (22.8;24.1) | 0.037 | <2.2e-16 | FPS > LFN > AHS |
| Intradaily variability (IV)[b] | 0.60 (0.48;0.72) | 0.63 (0.51;0.77) | 0.70 (0.54;0.85) | 0.037 | <2.2e-16 | LFN > FPS > AHS |
| Wake-Up Time | 6.9 (6.2;7.5) | 7.5 (6.6;8.1) | 6.9 (6.4;7.5) | 0.028 | <2.2e-16 | FPS > AHS,LFN |
| SD Wake-Up Time | 0.56 (0.37;0.80) | 0.72 (0.44;1.16) | 0.49 (0.31;0.77) | 0.027 | <2.2e-16 | FPS > AHS > LFN |
| Interdaily stability (IS)[b] | 0.76 (0.69;0.82) | 0.72 (0.64;0.79) | 0.73 (0.66;0.80) | 0.021 | 1.59e-13 | AHS > FPS,LFN |
| Mesor[a] | 2102 (1850;2359) | 2170 (1863;2433) | 2257 (1893;2678) | 0.020 | 2.0e-12 | LFN > FPS > AHS |
| Start of L5[b] | 0.23 (−0.70;1.13) | 0.73 (−0.38;1.97) | 0.55 (−0.27;1.35) | 0.016 | 1.59e-10 | FPS,LFN > AHS |
| Midpoint of L5[b] | 2.73 (1.80;3.63) | 3.23 (2.12;4.47) | 3.05 (2.23;3.85) | 0.016 | 1.59e-10 | FPS,LFN > AHS |
| Beta[a] | 8.03 (4.89;17.02) | 8.59 (5.02;20.14) | 12.33 (4.43;40.20) | 0.007 | 1.79e-05 | LFN > AHS,FPS |
| Other variables | | | | | | |
| Sleep Efficiency | 83 (77;87) | 64 (54;72) | 82 (75;87) | 0.218 | <2.2e-16 | AHS > LFN > FPS |
| Down-Mesor[a] | 22.0 (21.1;22.8) | 22.0 (21.0;23.1) | 19.9 (18.9;20.9) | 0.202 | <2.2e-16 | AHS,FPS > LFN |
| Sleep Maintenance | 87 (82;91) | 71 (62;79) | 85 (79;90) | 0.190 | <2.2e-16 | AHS > LFN > FPS |
| Relative amplitude (RA)[b] | 0.86 (0.82;0.90) | 0.77 (0.68;0.81) | 0.84 (0.78;0.88) | 0.169 | <2.2e-16 | AHS > LFN > FPS |
| Number of Naps | 2.5 (1.3;4.3) | 2.3 (1.1;3.8) | 5.5 (3.5;8.5) | 0.167 | <2.2e-16 | LFN > AHS,FPS |
| L5[b] | 292 (217;383) | 525 (414;682) | 323 (232;442) | 0.164 | <2.2e-16 | FPS > LFN > AHS |
| SD Midpoint (Onset interval) | 0.45 (0.30;0.63) | 0.75 (0.53;1.05) | 0.43 (0.27;0.68) | 0.089 | <2.2e-16 | FPS > AHS,LFN |
| Midpoint of M10[b] | 13.2 (12.3;14.2) | 13.7 (12.7;15.1) | 12.6 (11.9;13.6) | 0.039 | <2.2e-16 | FPS > AHS > LFN |
| Midpoint (Onset Interval) | 2.92 (2.37;3.48) | 3.48 (2.68;4.36) | 3.09 (2.57;3.65) | 0.038 | <2.2e-16 | FPS > LFN > AHS |
| Amplitude[a] | 3712 (3154;4280) | 3294 (2611;3894) | 3250 (2591;4125) | 0.038 | <2.2e-16 | AHS > FPS,LFN |
| Up-Mesor[a] | 6.9 (6.3;7.5) | 7.4 (6.6;8.1) | 7.0 (6.5;7.9) | 0.028 | <2.2e-16 | FPS > LFN > AHS |
| SD Midpoint (Bed interval) | 0.43 (0.30;0.61) | 0.58 (0.37;0.87) | 0.40 (0.25;0.63) | 0.026 | 3.06e-16 | FPS > AHS,LFN |
| M10[b] | 4049 (3543;4534) | 3884 (3370;4406) | 3736 (3111;4332) | 0.023 | 3.49e-14 | AHS > FPS > LFN |
| SD Bed Time | 0.51 (0.33;0.75) | 0.68 (0.41;1.01) | 0.51 (0.31;0.83) | 0.020 | 1.36e-12 | FPS > AHS,LFN |
| Bed Time | 22.7 (22.1;23.4) | 22.6 (21.8;23.5) | 23.0 (22.4;23.7) | 0.017 | 3.09e-11 | LFN > AHS,FPS |
| Midpoint (Bed Interval) | 2.79 (2.24;3.33) | 2.95 (2.30;3.73) | 2.97 (2.44;3.52) | 0.011 | 1.61e-07 | FPS,LFN > AHS |
| Time from Onset to Wake-Up | 460 (423;495) | 453 (387;508) | 444 (409;484) | 0.009 | 1.23e-06 | AHS > LFN |

Abbreviations: *AHS* Active healthy sleepers, *FPS* Fragmented poor sleepers, *LFN* Long and frequent nappers, *IQR* interquartile range. [a] computed from extended cosine model; [b] nonparametric measures; [c] Kruskal-Wallis test was used; [d] Dunn test adjusted for multiple comparisons using Bonferroni method was used.

AHS were characterized by normal nighttime sleep duration (median = 6.7 h), higher sleep quality (median sleep efficiency = 83%, sleep maintenance = 87%, minimum = 234, L5 = 292), earlier timing of sleep (median sleep onset time = 23.1, start and midpoint of L5 = 0.23 and 2.73, midpoint of bed and onset interval = 2.79 and 2.92), stronger circadian rhythmicity (median amplitude = 3712, pseudo-F = 1078, intradaily variability = 0.60, interdaily stability = 0.76, relative amplitude = 0.86), and higher activity during wake periods (median M10 = 4049, alpha = −0.40) (description and interpretation of all sleep/circadian data are described in Table 1).

FPS were characterized by shorter nighttime sleep duration (median = 5.6 h) and longer time in bed (median= 8.8 h), lower sleep quality (median sleep efficiency= 64%, sleep maintenance = 71), higher sleep fragmentation (median sleep latency = 53 min, wake after sleep onset= 126 min, L5 = 525, and median SD for sleep onset = 1.14, bedtime= 0.68, wake-up time = 0.72, midpoint of bed and onset interval = 0.58 and 0.75), later timing of sleep and activity (median acrophase = 14.69, wake-up time = 7.5, start of M10 = 8.7, up-mesor = 7.4, sleep onset time = 23.7, midpoint of onset interval = 3.48), and weaker circadian rhythmicity (median amplitude = 3294, relative amplitude = 0.77).

LFN were characterized by longer (median = 79 min) and more frequent naps (median = 5.5), normal nighttime sleep duration (median = 6.4 h), good sleep quality (median sleep efficiency = 82%, sleep maintenance = 85%), earlier timing of activity (median acrophase = 13.51, start and midpoint of M10 = 7.6 and 12.6, down-mesor = 19.9), and more fragmented circadian rhythmicity (median pseudo-F = 805, intradaily variability = 0.70, interdaily stability = 0.73, amplitude = 3250).

All sleep and circadian variables differed significantly across the three profiles ($p < 0.0001$). Among the cluster analysis variables, large effect sizes were found for the following (ordered by descending order of contribution): alpha ($\eta^2 = 0.213$), minimum ($\eta^2 = 0.193$), wake after sleep onset ($\eta^2 = 0.188$), minutes napping ($\eta^2 = 0.160$), and sleep latency ($\eta^2 = 0.157$). Other variables with large effect sizes included sleep efficiency, down-mesor, sleep maintenance, relative amplitude, number of naps, and L5 (Table 2). Sleep profiles based on the largest contributors were illustrated in Fig. 1.

Compared to AHS, FPS were more likely to live alone, to be less educated and less physically active, while LFN were slightly older. Both FPS and LFN were more likely to be non-White, smokers, to have a history of hypertension and a higher BMI. AHS consumed less caffeine than FPS (Table 3).

Exclusions of participants who intermittently used nightly mechanical devices during sleep ($n = 37$) showed similar sleep/circadian profiles (Supplementary Table 3).

### Dementia incidence
Among the 2562 men with dementia data, 461 (18.0%) incident dementia cases were identified (annual incidence rate = 27.6/1000 person-years) over 12 years of follow-up (median = 6.1 [IQR = 3.2-10.5]). Kaplan–Meier curves are shown in Fig. 2. In unadjusted models, FPS had an increased risk of dementia (hazard ratios (HR) = 1.34, 95% confidence intervals (CI) = 1.03-1.74) compared to AHS. There was no association with dementia risk for LFN (HR = 1.11, 95% CI = 0.89-1.39). After adjusting for demographics, behaviors, comorbidities and sleep medication use, results were similar (HR = 1.35, 95% CI = 1.02–1.78 for FPS and HR = 1.09, 95% CI = 0.86–1.38 for LFN). Sensitivity analyses adjusting for different set of covariates displayed comparable findings (Supplementary Tables 4–7) as well as the exclusion of incident dementia cases identified at the first follow-up visit (HR = 1.42, 95% CI = 1.01–1.98 for FPS and HR = 1.20, 95% CI = 0.90-1.61 for LFN; Supplementary Table 8).

### CVD event incidence
Among 2606 men with CVD data, 839 (32.2%) incident CVD events were identified (annual incidence rate = 42.4/1000 person-years) over 12 years of follow-up (median = 9.7 [IQR = 4.5-10.5]). Kaplan–Meier curves are shown in Fig. 2. In unadjusted models, both FPS and LFN were significantly associated with a higher risk of CVD events compared to AHS (HR = 1.44, 95% CI = 1.19–1.74 and HR = 1.21, 95% CI = 1.02–1.42, respectively). After multivariable adjustment, FPS were significantly associated with a higher risk of CVD events compared to AHS (HR = 1.32, 95% CI = 1.08-1.60), while LFN showed a borderline association (HR = 1.16, 95% CI = 0.98-1.37, $p = 0.08$). Results remained consistent in the sensitivity analysis after further adjustment (Supplementary Tables 4–6 and 9), although the association for LFN was strongly attenuated after exclusion of participants with a history of heart attack or stroke (HR = 1.07, 95% CI = 0.87-1.31; Supplementary Table 9).

### Discussion
In a prospective cohort of older men, we identified three distinct multidimensional sleep/circadian profiles using machine learning: active healthy sleepers [AHS], fragmented poor sleepers [FPS], and long and frequent nappers [LFN]. Compared to AHS, FPS had increased risks of developing dementia and CVD events over 12 years whereas LFN tended to have an increased risk of CVD events, but not dementia. These results suggest that poor sleep and disrupted circadian rhythms may be risk factors or preclinical markers of dementia and CVD and highlight potential target populations for sleep interventions.

Few studies have used clustering[32–34] or latent class[35–37] analyses to discern sleep profiles in older adults. Moreover, these studies faced important limitations, including cross-sectional design[32], reliance on self-reported sleep data[32,35,36], lack of rest-activity variables[32,34–36], and a focus on clinical populations[34]. To the best of our knowledge, this study is the first to identify objective sleep and circadian profiles in community-dwelling older men using both sleep and rest-activity parameters with prospective follow-up for health outcomes. We identified three sleep profiles with high heterogeneity, as evidence by the wide range of values and the significant differences in all actigraphy-derived variables across the groups ($p < 0.0001$ for all). The AHS group was the most common profile (64%), characterized by a combination of favorable characteristics: normal nighttime sleep duration, higher sleep quality, and stronger circadian rhythmicity. The LFN (21.9%) were characterized by longer and more frequent naps, alongside a combination of favorable and unfavorable dimensions: normal nighttime sleep duration, good sleep quality, and more fragmented circadian rhythms. The third group was the FPS (14.1%) who had a combination of unfavorable characteristics: shorter nighttime sleep duration, lower sleep quality, higher sleep fragmentation, delayed sleep/activity timing, and weaker circadian rhythmicity. Although each cluster is characterized by distinct main features, it is important to acknowledge that some overlap exists between the profiles. For instance, LFN's nighttime sleep efficiency and duration are comparable to those of AHS, suggesting shared good sleep quality between AHS and LFN, while FPS and LFN exhibit similar levels of circadian rhythmicity. This overlap indicates that the clusters represent a continuum of sleep and circadian behaviors, highlighting the complexity of sleep health. Compared to prior research, our study provides a deeper characterization of nighttime and daytime sleep patterns by using a broader set of objective parameters, including extensive analysis of circadian rhythms. This provided a more nuanced and complete understanding of participants' multidimensional sleep and circadian patterns. Additionally, the advanced machine learning technique has further enhanced classification accuracy.

Compared to AHS, FPS had a higher risk of dementia, consistent with variable-centered research linking short sleep duration, sleep fragmentation, poor sleep efficiency, and weak circadian rhythms with dementia incidence[2,6,38–40]. This result is also in line with our previous work demonstrating the association between a multidimensional measure of sleep health and long-term cognitive decline[41]. Our result extends those of a recent cross-sectional, person-centered study that used self-reported sleep, which found that the poor sleepers group performed worse on several cognitive tests compared to the healthy sleepers group[32]. Potential underlying mechanisms include accumulation of amyloid-beta and tau proteins, disturbed glymphatic clearance, metabolic dysfunction, inflammation, and disrupted 24-h melatonin rhythms[42–45]. Given

**Table 3 | Baseline characteristics according to identified sleep clusters among the 2,667 participants**

| | Active Healthy Sleepers (n = 1707) | Fragmented Poor Sleepers (n = 376) | Long and Frequent Nappers (n = 584) | | |
|---|---|---|---|---|---|
| Characteristics | Median (IQR) or No. (%) | Median (IQR) or No. (%) | Median (IQR) or No. (%) | p-value[a] | Post hoc[b] |
| Age (years) | 75 (71;79) | 76 (72;80.3) | 76 (72;81) | 0.002 | LFN > AHS |
| Education, ≤High school | 313 (18.3) | 92 (24.5) | 134 (22.9) | 0.005 | FPS > AHS |
| Race/ethnicity | | | | 0.0002 | |
| White | 1575 (92.3) | 327 (87.0) | 523 (89.6) | | |
| Black/African American | 36 (2.1) | 24 (6.4) | 22 (3.8) | | |
| Other | 96 (5.6) | 25 (6.6) | 39 (6.7) | | |
| Living alone | 191 (11.2) | 66 (17.6) | 62 (10.6) | 0.001 | FPS > AHS,LFN |
| PASE score | 145 (101;190) | 125 (84;176) | 138 (96;183) | 4.07e-05 | AHS > FPS |
| GDS score, ≥6 | 90 (5.3) | 28 (7.5) | 38 (6.5) | 0.20 | - |
| Smoking status | | | | 0.01 | |
| Never | 704 (41.3) | 125 (33.2) | 225 (38.5) | | |
| Past | 976 (57.2) | 238 (63.3) | 347 (59.4) | | |
| Current | 26 (1.5) | 13 (3.5) | 12 (2.1) | | |
| Caffeine intake (mg/day) | 184 (36;368) | 214 (48;405) | 136 (0;356) | 0.009 | FPS > LFN |
| Alcoholic drink per week, >1 | 937 (55.2) | 207 (55.5) | 308 (52.9) | 0.61 | - |
| Body mass index (kg/m²) | 26.6 (24.4;28.8) | 27.6 (25.3;30.7) | 27.0 (24.8;29.6) | 3.73e-08 | FPS, LFN > AHS |
| History of heart attack | 288 (16.9) | 67 (17.8) | 95 (16.3) | 0.82 | - |
| History of stroke | 56 (3.3) | 9 (2.4) | 28 (4.8) | 0.10 | - |
| History of diabetes mellitus | 207 (12.1) | 60 (16.0) | 85 (14.6) | 0.08 | - |
| History of hypertension | 802 (47.0) | 208 (55.3) | 314 (53.8) | 0.001 | FPS,LFN > AHS |
| Current sleep medication | 185 (10.8) | 57 (15.2) | 62 (10.6) | 0.05 | - |
| Antidepressants | 111 (6.5) | 37 (9.8) | 39 (6.7) | 0.07 | - |
| Benzodiazepine | 69 (4.0) | 16 (4.3) | 22 (3.8) | 0.93 | - |
| Other sleep medications | 32 (1.9) | 8 (2.1) | 10 (1.7) | 0.90 | - |
| AHI (n = 2,414) | 12.4 (6.1;23.2) | 18.1 (8.9,32.0) | 15.0 (6.4;28.2) | 6.05e-09 | FPS > LFN > AHS |

Abbreviations: *AHI* apnea-hypopnea index, *GDS* Geriatric Depression Scale, *IQR* interquartile range, *PASE* Physical Activity Scale for the Elderly.
[a] Kruskal-Wallis test was used for continuous variables, Chi-square test for categorical variables. [b] Dunn test adjusted for multiple comparisons using Bonferroni method was used for continuous variables, pairwise comparisons with Chi-square test adjusted for multiple comparisons using Bonferroni method was used for categorical variables.

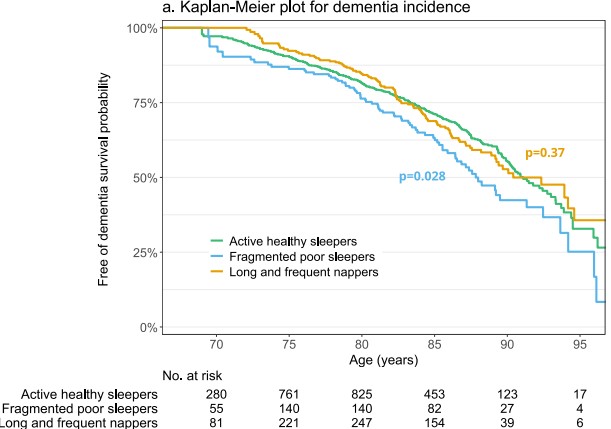
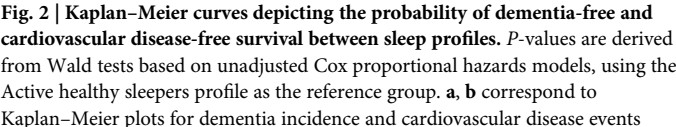
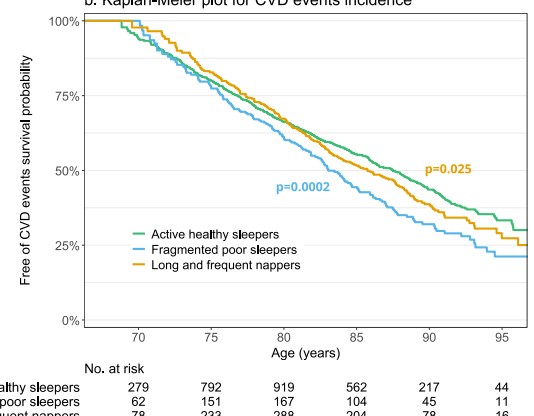

**Fig. 2 | Kaplan–Meier curves depicting the probability of dementia-free and cardiovascular disease-free survival between sleep profiles.** *P*-values are derived from Wald tests based on unadjusted Cox proportional hazards models, using the Active healthy sleepers profile as the reference group. **a, b** correspond to Kaplan–Meier plots for dementia incidence and cardiovascular disease events incidence, respectively. Green, blue, and yellow lines correspond to Active healthy sleepers, Fragmented poor sleepers, and Long and frequent nappers profiles, respectively. The sample size was *n* = 2562 for dementia outcome and *n* = 2606 for cardiovascular disease outcome. Abbreviations: CVD cardiovascular disease, No. number

melatonin's neuroprotective effects -such as reducing amyloid-beta production, mitigating tau hyperphosphorylation, alleviating oxidative stress, and enhancing blood-brain barrier function- its dysregulation may further link poor sleep health to increased dementia risk[45]. However, we cannot exclude the fact that preclinical dementia-related changes may also influence sleep and circadian patterns[46–48]. FPS also had an increased risk of CVD events, in line with several prior studies of individual sleep parameters[49–51]. Increase sympathetic activity and blood pressure, disrupted endothelial function, and inflammatory processes may explain in part this association[52]. Interestingly, AHI differed across the three groups, with the FPS group exhibiting the highest median AHI (Table 3). Adjusting for baseline AHI slightly attenuated the association between FPS and dementia risk (Supplementary Table 5), but the effect size and confidence intervals remained largely unchanged. This suggests that sleep apnea may partially mediate this relationship whereas the association between FPS and CVD remained robust after adjusting for AHI, highlighting the importance of other mechanisms. Taken together, these results showed that FPS were associated with poor incident cognitive and cardiovascular health.

We did not observe an association between LFN and dementia incidence. This finding contributes to the ongoing debate on napping and dementia. Some studies have reported that longer or more frequent naps were linked to a higher risk of dementia and faster cognitive decline[5,53], while others have found a lower risk[54,55] or no association[7,56]. Our study demonstrated that long and frequent napping, when combined with good nighttime sleep dimensions, might not affect the risk of dementia. This underscores the importance of clustering analysis and considering combination of sleep and circadian dimensions, as longer and more frequent naps alone were associated with a higher risk of dementia in our sample. Furthermore, this is in line with a previous clustering study which showed that a high sleep propensity group (characterized by long naps) was protective against all-cause mortality, while napping alone was associated with a higher risk[33]. Interestingly, LFN were linked to increased risk of CVD events, although the association was of marginal significance. Prior research on napping and CVD has been mixed, with several studies suggesting a higher risk of CVD associated with more frequent or longer naps[4,12], while others suggested a protective effect[13]. Daytime napping may result from short or poor nighttime sleep (as a compensatory mechanism) or indicate poor overall health, both of which can contribute to increase CVD risk. However, these hypotheses do not fully explain our findings since LFN had normal nighttime sleep duration with good sleep quality, and LFN did not differ from AHS regarding sociodemographic factors and comorbidities. Although the exact reasons why LFN might be associated with CVD but not dementia are not well-understood, assumptions include autonomic nervous system disruptions or other metabolic changes not examined in this study[57,58], which may impact more the cardiovascular risk. It may also involve cardiovascular mechanisms that do not relate to dementia risk or have a less direct effect on it. Further research, including mediation analyses, is needed to better understand the role of napping in relation to adverse health outcomes and their underlying mechanisms.

Our findings have important clinical and public health implications. By identifying common multidimensional sleep and circadian patterns in older men using advanced machine learning techniques, this study enhances our understanding of the interrelations between numerous sleep/circadian parameters and underscores the critical need for comprehensive sleep health assessment in clinical practice and research settings. Both FPS and LFN exhibited poor circadian activity rhythmicity, emphasizing the importance of this dimension of sleep health. Future studies should incorporate circadian rhythms when examining adverse outcomes. Moreover, our results highlight specific at-risk groups that could benefit from sleep interventions and prevention efforts, and support poor sleep patterns as a marker or risk factor for cognitive and cardiovascular health. Public health initiatives may consider prioritizing the screening and monitoring of older adults with weak circadian rhythms combined with poor nighttime sleep or with high daytime napping.

Strengths of this study include a 12 year longitudinal design with high retention rates, a multidimensional measure of sleep and rest-activity rhythms using objective measures, and consideration of numerous potential confounders. We also used an innovative machine learning approach capable of detecting clusters with flexible shapes, which standard clustering methods cannot achieve. However, there are also limitations. Although we applied a prespecified algorithm previously used in MrOS studies[5,23], which aligns with approaches from other cohort studies[11,59], the diagnosis of dementia relied on both objective (e.g., dementia medication, cognitive tests) and self-reported data (e.g., self-reported physician diagnosis), which may lead to outcome misclassification. Moreover, the timing of dementia incidence was based on study visit dates, which may not reflect the actual onset of dementia, and information on dementia subtypes was lacking. Due to the lack of biomarkers of neurodegeneration in this study, we were unable to determine whether specific sleep profiles are associated with distinct neurodegenerative pathways. Future research should incorporate objective biomarkers, such as phosphorylated tau (p-tau)217, p-tau181, or white matter hyperintensities, to better understand their relationship with sleep profiles. CVD events assessment relied in part on self-report potentially introducing bias, but this approach was completed with rigorous, objective assessments and expert confirmation, making it a well-established and widely accepted method in the literature[60,61]. Additionally, actigraphy data were available only at baseline, with a median duration of 5 nights. Future research should use longer actigraphy recordings to more robustly assess circadian rhythmicity and examine changes in sleep over time to identify dynamic profiles and minimize potential bias due to single-time-point assessments. This study predominantly involves White older men, limiting the generalizability of the results. Future research should replicate these methods in more diverse samples. Lastly, as an observational study, we cannot assume causal relationships between sleep profiles and dementia or CVD events.

## Conclusions

In older men, we identified three multidimensional actigraphy-derived sleep/circadian profiles. Compared to AHS, FPS were associated with less favorable cognitive and cardiovascular health over 12 years, while FPS were linked to increased risk of CVD events, but not dementia. These results suggest potential targets for sleep interventions and prevention efforts and emphasize the need for careful screening of poor sleepers for adverse outcomes. Moreover, our study highlights the importance of future research to consider combinations of sleep characteristics.

## Data availability

The data supporting the findings of this study are openly available at https://mrosonline.ucsf.edu[15,16]. The source data underlying Fig. 1 are provided in Supplementary Data 1, and those underlying Fig. 2 in Supplementary Data 2.

## Code availability

Code for identification of clusters is accessible at https://zenodo.org/records/15792521[62].

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

## Acknowledgements

The authors thank the study staff and all the men who participated in MrOS Sleep Study. K.Y. is supported in part by (NIA) R35AG071916 and R01AG066137. Y.L. is supported by the NIA grants R21AG085495 and R01AG083836. The MrOS Study is supported by National Institutes of Health funding. The following institutes provided support: the National Institute on Aging (NIA), the National Institute of Arthritis and Musculoskeletal and Skin Diseases (NIAMS), the National Center for Advancing Translational Sciences (NCATS), and NIH Roadmap for Medical Research under the following grant numbers: U01 AG027810, U01 AG042124, U01 AG042139, U01 AG042140, U01 AG042143, U01 AG042145, U01 AG042168, U01 AR066160, R01 AG066671, and UL1 TR002369). The National Heart, Lung, and Blood Institute (NHLBI) provided funding for the MrOS Sleep ancillary study "Outcomes of Sleep Disorders in Older Men" under the following grant numbers: R01 HL071194, R01 HL070848, R01 HL070847, R01 HL070842, R01 HL070841, R01 HL070837, R01 HL070838, and R01 HL070839.

## Author contributions

C.C. contributed to the conception and design of the work, the analysis, the interpretation of data, and the draft of the manuscript. M.W. contributed the design of the work and the analysis. Y.L. contributed to the interpretation of the data and the revision of the manuscript. K.L.S. and S.A.I. contributed to the revision of the manuscript. K.Y. contributed to the design of the work, the acquisition and interpretation of data, and the revision of the manuscript.

## Competing interests

The authors declare no competing interests.
