## [Transparent Peer Review file · Communications Medicine]

Multidimensional Sleep Profiles via Machine learning and Risk of Dementia and Cardiovascular Disease

Corresponding Author: Dr Clémence Cavallès

Version 0:

Reviewer comments:

Reviewer #1

(Remarks to the Author)

This is a revised version of the paper by Cavallès et al. that I had the pleasure to previously review. I would like to thank the authors for carefully addressing all my comments. I greatly appreciated supplementary analyses confirming that sleep profiles remain consistent when excluding participants with mechanical device use, and that the potential influence of sleep apnea was further explored and discussed. I have no remaining points at this stage, except that I would advise mentioning in the limitations section that the duration of actigraphy recordings was relatively short to robustly measure circadian rhythmicity. Overall, I think that this paper is an important addition for the field.

Reviewer #4

(Remarks to the Author)

The methodology presented in this article is highly innovative. Unlike prior studies that examined the association between sleep metrics and CVD or dementia events independently, this study integrates multiple sleep indicators and employs clustering techniques to classify the population into three distinct sleep profiles: Active Healthy Sleepers (AHS), Fragmented Poor Sleepers (FPS), and Long and Frequent Nappers (LFP). This approach not only comprehensively evaluates the interdependencies among various sleep indicators but also underscores the notion that "sleep" cannot be adequately categorized based on a single or limited set of indicators, thereby highlighting the inherent complexity of the sleep process. Statistical analysis reveals significant differences in the characteristics of these three sleep profiles, while also identifying some overlap between them, which also reflects the complexity of sleep. Furthermore, survival analysis demonstrates markedly different incidence rates of dementia and CVD events across the three groups, providing deeper insights into the relationship between sleep patterns and both cognitive and cardiovascular health. Despite its strengths, I still have some major problems.

1. As the author stated, "Research has primarily examined sleep characteristics in isolation..." (line 43, page 3), a comparison between the "isolated sleep characteristics" and the clustering method employed in this study should be included to further substantiate the author's perspective. For instance, clustering based on a single sleep characteristic failed to adequately separate the Kaplan-Meier (KM) curves, and so on.
2. The stability of the clustering results must be validated. For example, some subjects could be randomly excluded, and the clustering process repeated multiple times to assess consistency. Additionally, beyond the MrOS dataset, both MESA and SHHS datasets contain sleep data and CVD events that could be utilized to validate the clustering results.
3. Supplementary Tables 3-8 should not remain exclusively in the supplementary materials. Key information from these tables should be extracted and incorporated into the main manuscript to enhance clarity and accessibility.
4. The order of sleep characteristics presented in Tables 2 and 3 should be adjusted to better emphasize the differences among the three groups.
5. The font size in Figure 2 is too small, and the p-values for the Kaplan-Meier (KM) analysis should be clearly indicated.

Reviewer #5

(Remarks to the Author)

The manuscript presents a rigorous and well-conducted analysis of objectively measured sleep and circadian behavior in a large cohort of older men. The use of principal component analysis and MCGHD mixture modeling to identify distinct sleep/circadian profiles is statistically robust and well-justified, particularly given the skewed distribution of actigraphy variables. The three identified profiles are characterized and clinically meaningful. Importantly, the associations with incident dementia and cardiovascular events over a 12-year follow-up are compelling and of significant public health interest. The inclusion of sensitivity analyses strengthens the credibility of the findings.

One suggestion for improvement would be to provide a brief explanation or visualization of PCA loadings to improve transparency around the features contributing most to the derived components. Additionally, while the rationale for using eta-squared effect sizes is appropriate, highlighting more clearly which specific variables (beyond alpha and minimum) most differentiated the clusters could further aid interpretability.

Overall, this high-quality contribution offers novel insights into the heterogeneity of sleep and circadian patterns in aging and their long-term health implications.

Version 1:

Reviewer comments:

Reviewer #4

(Remarks to the Author)

The authors have addressed all my comments well, acceptance recommended.

Reviewer #5

(Remarks to the Author)

Thank you for thoroughly addressing all the points raised in the previous round. I have no further comments or requests at this time.

Reviewers' comments:

Reviewer #1 (Remarks to the Author):

This is a revised version of the paper by Cavallès et al. that I had the pleasure to previously review. I would like to thank the authors for carefully addressing all my comments. I greatly appreciated supplementary analyses confirming that sleep profiles remain consistent when excluding participants with mechanical device use, and that the potential influence of sleep apnea was further explored and discussed. I have no remaining points at this stage, except that I would advise mentioning in the limitations section that the duration of actigraphy recordings was relatively short to robustly measure circadian rhythmicity. Overall, I think that this paper is an important addition for the field.

We strongly thank the reviewer for his/her comments. We added a point in the Limitations section noting that the duration of actigraphy recordings was relatively short to robustly measure circadian rhythmicity: “Additionally, actigraphy data were available only at baseline, with a median duration of 5 nights. Future research should use longer actigraphy recordings to more robustly assess circadian rhythmicity and examine changes in sleep over time to identify dynamic profiles and minimize potential bias due to single-time-point assessments” (pages 17-18).

In addition to my own review, I was asked to check whether referee #2's comments were addressed by the authors. In my opinion, all concerns of referee #2 were properly addressed. The authors thoroughly clarified the methodology, and added supplementary analyses which all confirmed initial results. I think that the paper is a very important addition to the field.

We appreciate the reviewer comment.

Reviewer #4 (Remarks to the Author):

The methodology presented in this article is highly innovative. Unlike prior studies that examined the association between sleep metrics and CVD or dementia events independently, this study integrates multiple sleep indicators and employs clustering techniques to classify the population into three distinct sleep profiles: Active Healthy Sleepers (AHS), Fragmented Poor Sleepers (FPS), and Long and Frequent Nappers (LFP). This approach not only comprehensively evaluates the interdependencies among various sleep indicators but also underscores the notion that "sleep" cannot be adequately categorized based on a single or limited set of indicators, thereby highlighting the inherent complexity of the sleep process. Statistical analysis reveals significant differences in the characteristics of these three sleep profiles, while also identifying some overlap between them, which also reflects the complexity of sleep. Furthermore, survival analysis demonstrates markedly different incidence rates of dementia and CVD events across the three groups, providing deeper insights into the relationship between sleep patterns and both cognitive and cardiovascular health. Despite its strengths, I still have some major problems.

1. As the author stated, "Research has primarily examined sleep characteristics in isolation..." (line 43, page 3), a comparison between the "isolated sleep characteristics" and the clustering method employed in this study should be included to further substantiate the author's perspective. For instance, clustering based on a single sleep characteristic failed to adequately separate the Kaplan-Meier (KM) curves, and so on.

As recommended by the reviewer, we have added a sentence in the Introduction highlighting key methodological advantages of our approach compared to analyzing sleep characteristics in isolation: “Moreover, such approach offers key methodological advantages over analyzing sleep characteristics in isolation, such as capturing interactions between multiple sleep dimensions and improving differentiation of outcome risks by leveraging more homogenous groups.” (page 4).

2. The stability of the clustering results must be validated. For example, some subjects could be randomly excluded, and the clustering process repeated multiple times to assess consistency. Additionally, beyond the MrOS dataset, both MESA and SHHS datasets contain sleep data and CVD events that could be utilized to validate the clustering results.

We thank the reviewer for his/her pertinent comment. As recommended by the reviewer, we checked the stability of our clustering results by randomly excluding 5% of our sample over 100 repetitions. Then, we compared the clusters obtained in the subsample to the ones obtained in the sample and analyzed the Adjusted Rand Index (ARI). This measure evaluates the level of agreement between two clusterings and ranges from -1 (less agreement than expected by chance between two clusterings) to 1 (perfect agreement) with 0 suggesting random agreement. We obtained a mean ARI of 0.58, a median of 0.55, a Q1 of 0.50 and a Q3 of 0.74, suggesting moderate to good clustering quality. Moreover, half of the ARI values fall between 0.50 and 0.74 showing reasonably consistent clustering. We added this point in the Methods section: “To assess the stability of the clustering solution, we applied a subsampling approach in which 5% of participants were randomly excluded, and the clustering algorithm was rerun on each subsample (n=100). The similarity between each subsample clustering and the original full-sample clustering was quantified using the Adjusted Rand Index (ARI), which ranges from -1

(less agreement than expected by chance between two clusterings) to 1 (perfect agreement) with 0 indicating random agreement.” (pages 8-9) and in the Results section: “The stability of the clustering results, as assessed by the ARI, indicated moderate to good quality (mean ARI= 0.58, median= 0.55, Q1= 0.50, Q3= 0.74).” (page 10).

While SHHS and MESA include sleep-related data, they do not offer the full range of actigraphy-derived measures—capturing sleep, daytime activity, and rest-activity rhythms—that underpin our clustering approach. SHHS, in particular, appears to include only PSG data. Additionally, the small number of male participants in comparable subsamples further limits the feasibility of external validation. Given these constraints, we focused on internal validation using robust subsampling techniques to assess the stability of our clustering results.

3. Supplementary Tables 3-8 should not remain exclusively in the supplementary materials. Key information from these tables should be extracted and incorporated into the main manuscript to enhance clarity and accessibility.

We agree with the reviewer and we added key results of these tables into the main manuscript: “Sensitivity analyses adjusting for different set of covariates displayed comparable findings (new Supplementary Tables 4-7) as well as the exclusion of incident dementia cases identified at the first follow-up visit (HR=1.42, 95% CI=1.01-1.98 for FPS and HR=1.20, 95% CI=0.90-1.61 for LFN; new Supplementary Table 8).” (page 12) and “Results remained consistent in the sensitivity analysis after further adjustment (new Supplementary Tables 4-6 and 9), although the association for LFN was strongly attenuated after exclusion participants with a history of heart attack or stroke (HR=1.07, 95% CI=0.87-1.31; new Supplementary Table 9).” (page 13).

4. The order of sleep characteristics presented in Tables 2 and 3 should be adjusted to better emphasize the differences among the three groups.

We have revised the presentation of sleep characteristics in Tables 1 and 2 (Table 3 does not include sleep characteristics). In Table 2, we have organized the variables based on effect size (η^2), in descending order (pages 32-33). This approach highlights the most discriminative variables across the three groups such as alpha, minimum and wake after sleep onset, thereby improving clarity and interpretability. In Table 1, which provides definitions of the sleep variables, we adopted an alphabetical ordering to improve accessibility and ease of reference for the reader (pages 30-31). Finally, Table 3, which presents baseline covariables unrelated to the sleep clusters, has been retained in a conventional order to maintain consistency and readability: sociodemographic followed by health-related variables.

5. The font size in Figure 2 is too small, and the p-values for the Kaplan-Meier (KM) analysis should be clearly indicated.

We increased the font size in Figure 2 and we added the p-values for the Kaplan-Meier analysis.

Reviewer #5 (Remarks to the Author):

The manuscript presents a rigorous and well-conducted analysis of objectively measured sleep and circadian behavior in a large cohort of older men. The use of principal component analysis and MCGHD mixture modeling to identify distinct sleep/circadian profiles is statistically robust and well-justified, particularly given the skewed distribution of actigraphy variables. The three identified profiles are characterized and clinically meaningful. Importantly, the associations with incident dementia and cardiovascular events over a 12-year follow-up are compelling and of significant public health interest. The inclusion of sensitivity analyses strengthens the credibility of the findings.

- 1. One suggestion for improvement would be to provide a brief explanation or visualization of PCA loadings to improve transparency around the features contributing most to the derived components.**

We agree with the reviewer and we added a description of the loadings for each variable used to create the clusters for each of the 6 principal components selected (new Supplementary Table 2). We added this point in the Results section: “The PCA loadings for each variable across PCs are presented in Supplementary Table 2.” (page 10) and in the Supplementary Materials document (new Supplementary Table 2).

- 2. Additionally, while the rationale for using eta-squared effect sizes is appropriate, highlighting more clearly which specific variables (beyond alpha and minimum) most differentiated the clusters could further aid interpretability. Overall, this high-quality contribution offers novel insights into the heterogeneity of sleep and circadian patterns in aging and their long-term health implications.**

We thank the reviewer for his/her comment. We have organized the variables in Table 2 based on effect size (η^2), in descending order (pages 32-33). This approach highlights the most discriminative variables across the three groups such as alpha, minimum and wake after sleep onset, thereby improving clarity and interpretability. Moreover, we modified the sentences within the text to highlight more clearly the largest contributors: “Among the cluster analysis variables, large effect sizes were found for the following (ordered by descending order of contribution): alpha ($\eta^2=0.213$), minimum ($\eta^2=0.193$), wake after sleep onset ($\eta^2=0.188$), minutes napping ($\eta^2=0.160$), and sleep latency ($\eta^2=0.157$). Other variables with large effect sizes included sleep efficiency, down-mesor, sleep maintenance, relative amplitude, number of naps, and L5 (Table

2).” (pages 11-12). Also, Figure 1 described the different sleep profiles based on the largest contributors.

Upon reviewing the initial concerns raised by referee #3 and the authors’ detailed responses, I believe the authors have adequately addressed the reviewer’s points.

Thank you for acknowledging this point.

REVIEWERS' COMMENTS:

Reviewer #4 (Remarks to the Author):

The authors have addressed all my comments well, acceptance recommended.

Reviewer #5 (Remarks to the Author):

Thank you for thoroughly addressing all the points raised in the previous round. I have no further comments or requests at this time.

We thank the reviewers for recommended acceptance of the manuscript.